# Attitudes toward Conservation of the Transboundary Białowieża Forest among Ecotourism Businesses in Poland and Belarus

**Marek Giergiczny** [1], **Sviataslau Valasiuk** [1], **Mikita Yakubouski** [2], **Mikołaj Kowalewski** [1,3], **Jędrzej Maskiewicz** [1] **and Per Angelstam** [4,5,*]

1    Faculty of Economic Sciences, University of Warsaw, Długa 44/50, 00-241 Warszawa, Poland; mgiergiczny@wne.uw.edu.pl (M.G.); svalasiuk@wne.uw.edu.pl (S.V.); mk.kowalewski@student.uw.edu.pl or dcb581@alumni.ku.dk (M.K.)
2    State Nature Protection Institution "National Park "Biełavieskaja Pušča", a/h Kamianiuki, 225063 Kamianiecki Rajon, Belarus; nik1989384@mail.ru
3    Department of Food and Resource Economics, University of Copenhagen, Rolighedsvej, 23 DK-1958 Frederiksberg C, Denmark
4    Department of Forestry and Wildlife Management, Inland Norway University of Applied Sciences, Campus Evenstad, N-2480 Koppang, Norway
5    Faculty of Forest Sciences, School for Forest Management, Swedish University of Agricultural Sciences, SE-730 21 Skinnskatteberg, Sweden
*    Correspondence: per.angelstam@inn.no or per.angelstam@slu.se; Tel.: +46-702444971

**Abstract:** The Białowieża Forest is a contested transboundary forest massif in Poland and Belarus. Reflecting on transitions from value chains built on sustained yield forestry to ecotourism, we pioneer documentation of how country-specific legacies shape preferences toward increased forest protection at the expense of wood production. For both countries, we used a quantitative ordered logit model based on questionnaires to Polish and Belarusian ecotourism business owners to, for the first time, empirically study drivers of their preferences toward different Białowieża Forest values, and we used qualitative data to identify attitudes toward the expansion of protected areas in the Białowieża Forest. Whilst Belarusian ecotourism business owners supported increased area protection, the opposite was true for their Polish counterparts. The proportion of foreign guests co-varied with support toward increased area protection. Conversely, local origin, size of hospitality business, and role of foresters as customers decreased interest in area protection. The qualitative data revealed that narratives against extended area protection were spread in Poland but not in Belarus. The conflict over the conservation of the Polish part of the Białowieża Forest involves actors and stakeholders with competing interests. A solution is that this remnant massif of the once widespread European temperate lowland forest becomes subject to a regional planning and zoning perspective. Encouraging multiple value chains and evidence-based collaborative learning are key components.

**Keywords:** Białowieża National Park; biodiversity conservation conflict; forest management; nature-based tourism; nature protection; ordered logit model; Polish State Forest Holding

## 1. Introduction

In Europe, remnants of naturally dynamic forests and landscapes are extremely scarce [1,2]. Ambitions to improve opportunities for biodiversity conservation have triggered the development of policies aiming at increasing the number of protected areas [3]. Extending existing protected areas, often combined with nature restoration through the rewilding of once-degraded landscapes, is thus widely seen as a means for coping with the human footprint on nature [4]. European countries in the East are typically less impacted by human endeavor than in the West (e.g., [5]), which offers opportunities to protect and conserve existing remnants of functionally connected habitat networks. However, efforts to

expand protected areas have led to conflicts. As a contiguous forest massif on both sides of the border between Poland and Belarus, the Białowieża Forest is a prime example [6].

In Central and Eastern Europe, approaches for extension of protected areas reflect a mix of systemic post-Soviet transformation on the one hand, and infrastructural, institutional and cultural Europeanization on the other e.g., [7–10]. Giving local communities a say in debates over the designation or extension of existing protected areas is a normal procedure in modern Europe, sometimes leading to serious impediments to increasing the number of protected areas for biodiversity conservation.

For example, in Poland, the early period of systemic post-Soviet transformation, when top-down decision-making was the norm, coincided with a "golden era" of increasing areas aimed at nature protection. By 2001, the overall number of national parks had increased from 14 to 23 [11]. This trend was brought to a halt after Poland passed a new conservation law in 2001 requiring the acceptance of local communities. As a result, no other parks have been established since then.

Expansion of existing protected areas is commonly associated with conflicts among actor and stakeholder groups. Motivations for conflicts may emanate from socioeconomic, political, and cultural phenomena such as land ownership and access rights; environmental and social justice and civil rights; ways of knowing nature and ideologies; as well as particular management practices [12–14]. The essence of the conflicts typically comprises restrictions to human activities [12–14], limited access to ecosystem services [12–14] and increased natural and anthropogenic disturbance [12–14]; increase in bureaucratic procedures [10,14]; shifts in institutional roles and property rights [14] or in cultural identity [15]; and asymmetry of information/perception [14]. Negative consequences of conflicts about area protection range from the displacement of people [16,17] to intangible phenomena such as violated place identity [18].

Positive attitudes in local communities toward the establishment or enlargement of protected forest areas are related to the opportunity for benefits from different value chains that people derive directly or indirectly from protected areas [19–22]. At the same time, for value chains linked to forestry for wood production, expanding area protection and conservation may entail real costs and lost income, or fear thereof. This is complicated by the spatial scales at which benefits and costs occur. The conflict around the protection of the Białowieża Forest is a classic example of conflict about the provision of public goods. Here, evidence (e.g., [23–27]) points to national benefits exceeding local costs. However, these benefits contrast with reluctance among local communities arising from perceived or actual local costs exceeding local benefits (see e.g., [28–32]).

Interestingly, households located within the Białowieża Forest massif are reluctant toward increasing area protection [32], and simultaneously, those benefiting most from nature-based tourism [31–33]. For instance, more than twice as many local citizens work in the new and growing tourism industry than in the old declining forestry sector [33,34]. This makes ecotourism business owners' preferences extremely important amidst the debate about stewardship and conservation of the Białowieża Forest massif. Moreover, owners of the agro-tourism farmsteads located inside the Białowieża Forest massif or in its immediate outskirts might be assumed economically incentivized to support the expansion of area protection because their location provides the opportunity to increase their income by charging premium prices for their services [35]. The literature on environmental attitudes of touristic industry businesses owners (e.g., [36–40]) including local B&B farmstead owners (e.g., [41]), as well as the touristic business owners operating in the vicinity of protected areas (e.g., [42–44]), mostly concentrates on environmental aspects of internal hotel management issues (e.g., energy, water use or waste treatment).

The Białowieża Forest is divided into a Polish and a Belarusian part. In contrast to previous studies, the aim of this study is to pioneer the empirical examination of how country-specific socioeconomic and power contexts shape the preferences of owners of agro-tourism businesses toward increased area protection. Combining quantitative and qualitative survey data, we identify perceived winners and losers from improved

biodiversity conservation amidst the long-lasting conflict over the extension of protected areas in the transboundary Białowieża Forest massif. The border between these two countries can be viewed as an immaterial fault line [45] represented by the European Union (Poland), and the Soviet legacies of the union between Belarus and the Russian Federation. The western and eastern parts of the Białowieża Forest massif can thus be viewed as a "natural experiment" [46] ideally suited to address tourism businesses' preferences toward increased nature protection in their immediate neighborhood.

## 2. Methodology

### 2.1. Overview of Research Approach

The portfolios of forest values and their role in value chains supporting rural livelihoods are dynamic in time and space [47]. For example, in the past, multiple uses of forest landscapes were the foundation of traditional village systems [48]. The industrial revolution and the associated growth of transport infrastructure for bulky products coincided with an increased role of wood, the production of which became a key mission for forestry. Since the 1990s, Sustainable Forest Management, stressing the roles of satisfying economic, ecological, and social values and inclusive governance, has emerged. This has triggered debate and conflict (e.g., [6]) over how to accommodate multiple forest values.

Landscapes with different histories and systems of governance can be used as a "natural experiment" [49] using a case study approach. Following the terminology of [50,51] the unit of study is a "bounded" separate entity in terms of place and physical boundaries hosting a neighborhood, organizations, or cultures. With a single place-based case study approach, one can carry out an in-depth exploration of a specific bounded system. In this study, we use the transboundary Białowieża Forest, located across the border between an EU country (Poland) and a country that has kept legacies of the former USSR (Belarus), as a place-based case study. Viewing the Białowieża Forest as a social-ecological system, we (1) address desires to protect more forests as an asset for the hospitality industry and nature-based tourism in different systems of societal steering. (2) We present the case study of ecological and social systems to review the zoning approach in the transboundary Białowieża Forest massif in Poland and Belarus to maintain different forest values as a base for rural development, and the conflicts among actors and stakeholders around this. We then (3) collected empirical data on the attitudes of agro-tourism lodge owners as a novel value chain using quantitative and qualitative methods, and analyze them. Finally, (4) we discuss similarities and differences between Poland and Belarus (i.e., negative attitudes toward greater protection in Poland while positive in Belarus), and propose how zoning and new modes of governance and regional planning could sustain a transition to a multifunction landscape.

### 2.2. Case Study: The Cross-Border Białowieża Forest Massif in Poland and Belarus

#### 2.2.1. The Ecological System: A Unique Forest Remnant and Zoning of Functions

The potential natural vegetation in most of the European continent, except drylands and at high altitudes and latitudes, is forest and woodland. However, transformation to agricultural land has reduced forest cover beyond critical thresholds for the conservation of specialized forest species (e.g., [52], Figure 1). Due to a very particular history [47], this study focuses on the Białowieża Forest as an individual unique remnant fragment of the once contiguous lowland temperate forest in Central Europe. Historically, centuries of providing local livelihoods, and as a guarded game ground for the ruling classes in the Grand Duchy of Lithuania, Poland, and Russia [53] predate more than a century of timber harvesting, which started during World War I.

The 20th century brought about many changes in the Białowieża Forest massif [50]. During World War I, many prisoners of war were engaged in logging, some of whom remained in Białowieża and created a new class of forest and industry workers. After World War II, the Belarusian and Polish parts of the Białowieża Forest massif were divided. In Belarus, but not in Poland, collectivization of land use took place. In recent decades the

Polish part of the Białowieża Forest massif rural areas are becoming depopulated due to migration to cities or abroad.

The Białowieża Forest is a forest biodiversity hotspot with virtually complete sets of large herbivores and top predators, as well as old-growth forest specialists (e.g., woodpeckers, saproxylic insects, bracket fungi, lichens, and mosses) [54]. A unique feature of the Białowieża Forest is that natural and close-to-natural ecological processes are still operating at a landscape scale (e.g., large-scale trophic interactions; [55]). In this study, we use the term Białowieża Forest for the entire transboundary forest massif located in Poland and Belarus (Figure 1).

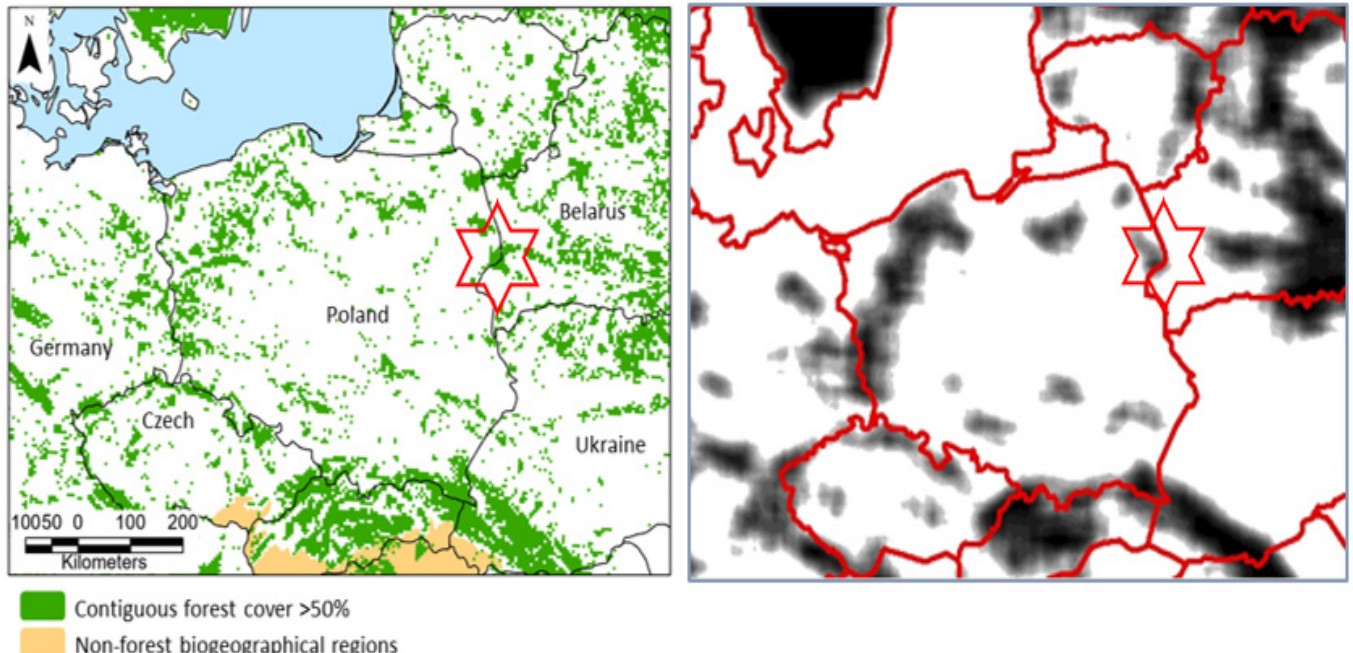

Contiguous forest cover >50%
Non-forest biogeographical regions

**Figure 1.** Central European countries such as Poland (PL) and Belarus (BY) have lost much of their natural potential forest areas to other land covers (white on the left map). As a consequence, there are only a few remnant forest massifs as isolated islands with high forest cover (left; own GIS analysis of data in [56]) and with landscape-level functional connectivity (right; redrawn using Figure 2 in [5]), generally confined to mountain regions, inaccessible wetlands and to poorer soils. The Białowieża Forest (marked with a star) is a rare exception.

The Białowieża Forest massif forms a contiguous forest-dominated area covering ~1500 km$^2$ in Poland and Belarus. Research in multiple disciplines indicates that during the past two millennia, this area has been subject to almost continuous anthropogenic interventions of varying, though mostly low, intensity in time and space, ranging from animal husbandry and wood harvesting to royal and imperial hunting reserves [47]. Today, the Białowieża Forest is covered by a range of forest types and management histories, from some being modified by forestry for wood production to others being close to natural forest [54]. The variation of parameter values for indicators of forest naturalness, such as dead wood, large trees and older stands, and associated specialized species, matches the gradient among zoning categories of the Białowieża Forest [57–59]). The first protected area was established in 1921 under Polish jurisdiction and was later transformed into a de facto national park in 1932 as a part of the state forests, and an independent national park in 1947. Since 1944 the Białowieża Forest massif has been divided into a Polish (about 1/3) and a (Soviet) Belarusian (the remaining 2/3) part. A semi-intact naturally dynamic part of the Białowieża Forest massif is currently protected in two adjacent national parks in Poland and Belarus.

Today, the Białowieża Forest massif has a wide range of international and national designations with different legal definitions, management, and governance, and they belong to Poland, Belarus, or are transboundary. This includes the transboundary UNESCO World Heritage site, two UNESCO MaB International Biosphere Reserves, national parks, nature reserves, Natura 2000 integrated area for the special protection of birds and habitats, Emerald Network site, wetland of international importance established under the auspices of the Ramsar convention, three Important Bird Areas, Protected Landscape Area, and a State Forest's Promotional Forest Complex. Taken together, this forms a spatial zoning system with several coarse categories in both Poland and Belarus, which range from strict protection, allowing natural processes via active conservation management, to timber-oriented forest management (Table 1).

**Table 1.** Hierarchy of denominations for different zones in the Białowieża Forest massif, see also Figure S1 [60].

| | Transboundary Białowieża Forest Massif<br>Białowieża Biosphere Reserves in Poland and Belarus<br>World Heritage Site (WHS) + Buffer Zone | |
| --- | --- | --- |
| | Poland | Belarus |
| Functional categories for zoning | National terminology | National terminology |
| Strict protection<br>(limited access) | Białowieski NP strictly protected area | Strict control zone |
| Conservation management<br>(limited access) | Białowieski NP active conservation management zone<br>and nature reserves | Controlled use zone (plus parts of strict control zone) |
| Managed forest—without wood removal<br>(free access, no forest management) | Managed State Forest Holding | Recreational zone |
| Managed forest—with removal allowed<br>(free access, silviculture aimed at re-naturalization)<br>Active protection of biodiversity and landscape | Managed State Forest Holding | Business zone with economic activities (inside and outside WHS) |

In both the Polish and Belarusian parts of the Białowieża Forest massif, there are near-natural dynamic forest areas with strict protection within national parks, which are subject to a total ban on human interference with the natural ecosystems and processes. The strictly protected core areas are surrounded by forests where economic activities are allowed to various extents, depending on a particular zone's regulations.

The next functional zoning category includes the active conservation management zone of the Białowieski NP in Poland and a suite of nature reserves scattered amongst the managed forests. In Belarus, this corresponds to the controlled use zone.

Finally, there are two categories of managed forests belonging to the Polish State Forest, with and without wood removal being allowed. According to the agreement with UNESCO, wood harvesting is allowed in the latter zone, only if it is aimed at enhancing biodiversity. In Belarus, there is a business zone with wood harvesting, hunting, and paying guests, and a recreation zone. The areas and area proportions of different categories of zoning are summarized in Figure 2, and detailed in Table 2.

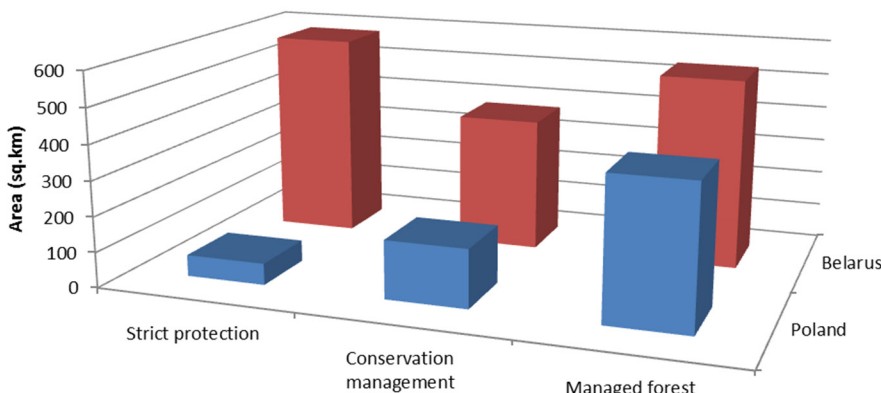

**Figure 2.** Distribution in Poland and Belarus of strictly protected areas, areas with conservation management, and managed forests.

**Table 2.** Zoning purposes and national terminology for different categories, total areas, and proportions in the Polish and Belarusian parts of the Białowieża Forest World Heritage Site, and in the entire transboundary Białowieża Forest massif.

| Functional Categories for Zoning | Poland (Adapted from [54]) | | | Belarus [61] | | | Transboundary Białowieża Forest Massif [60] | |
|---|---|---|---|---|---|---|---|---|
| | Land Users/Categories | Area, km² | % of the Total | Land Users/Categories | Area, km² | % of the Total | Area, km² | % of the Total |
| Strict protection | Białowieski NP strictly protected | 60.6 | 9.7 | Strict control zone | 583.0 | 38.8 | 643.6 | 30.3 |
| Conservation management | Białowieski NP active conservation management zone | 44.6 | 7.1 | Controlled use zone | 379.1 | 25.3 | 544 | 25.6 |
| | Nature reserves | 120.3 | 19.2 | | | | | |
| Managed forest (without or with logging) | Managed State Forest Holding | 399.5 | 63.9 | Recreation zone | 80.1 | 5.3 | 938.3 | 44.1 |
| | | | | Business zone | 458.7 | 30.6 | | |
| TOTAL | | 625 | 100 | | 1500 | 100 | 2125 | 100 |

The national parks in Poland and Belarus differ substantially with respect to their governance as well as to opportunities for potential extension. In Poland, being a western liberal democracy and EU member, designation and/or extension of any national park, being a procedure supervised by the country's Ministry for Environment, can be legally blocked at the municipality level. As a result, despite numerous attempts, since 2001, no significant changes occurred in regard to the enlargement of the existing Polish Białowieski NP (16% of the Białowieża Forest massif's Polish part). In contrast, in the post-Soviet highly centralized Belarusian national park concept, the administration is directly subordinated to the presidential affairs office, and processes of designation and extension follow top-down procedures. Thus, in Belarus, the NP "Biełavieskaja Pušča" has been enlarged twice since its establishment in 1991, and its strictly protected zone has been extended three times during the same period. However, we are not aware of any independent analyses of the consequences of different management practices on species, forest structure, and ecological processes. Regarding the expansion of the protected areas into the managed forest zone, a key topic is whether or not to expand the similarly sized zones allowing logging (zone 4) at the expense of the zone without logging (zone 3).

The multitude of various competencies and authorizing institutions often interfere with each other, thus impeding effective conservation and use of the transboundary Białowieża Forest massif [32,62]. Despite the internationally adopted recommendations on transboundary and the coordination of the conservational effort over the Białowieża Forest massif [63,64], there has only been limited coordination of activities across the state border [27,65].

2.2.2. The Social System: Disputes over the Białowieża Forest Massif

The societal conflict over forestry aimed at wood production vs. biodiversity conservation concerning the transboundary Białowieża Forest massif has continued for a century [54], and more recently, since the beginning of the systemic post-soviet transformation in Poland and Belarus in the late 1980s and early 1990s. In Poland, the conflict between local authorities and State Forest Holding, on the one hand, vs. the conservationist community on the other, has a high profile in the literature as well as in the public discourse (e.g., [6,34,54,65–70]). In contrast, even if the conflict in Belarus is less spectacular, both conflicts have much in common. However, the Belarusian situation has not been subject to review.

This motivates a comprehensive introduction to the Belarusian part of the Białowieża Forest in this section. Both the most recent iteration of the conflict in Poland in 2016–2018 and an earlier 2002–2008 conflict in Belarus were provoked by bark beetle (*Ips typographus* L.) infestations. This is an insect attacking Norway spruce (*Picea abies* (L.) H. Karst) stands, and may lead to mass die-off [71]. As a response, foresters and their allies undertook campaigns of fighting the bark beetle through extended logging, including in nature reserves, to prevent further expansion of infestations, and "save" the forest's wood value. At the same time, their opponents insisted on conserving the naturalness and epidemic character of the bark beetle outbreaks caused by a combination of climate change, lowered groundwater table, and concentration of older tree stands in the Białowieża Forest massif [72]. This triggered a dispute between the conservation community, which advocated covering the entire Polish part by the national park regime, and/or expanding the strict control zone of the national park, over the entire Belarusian segment of Białowieża Forest's massif, thus imposing permanent restrictions on forestry activities.

Thus, bark beetle infestations served as the *casus belli* root cause of a much older and more general dispute linked to the choice of management paradigm appropriate for sustaining biodiversity conservation values. Indeed, the conflicts in the Polish and Belarusian segments of the Białowieża Forest massif are manifestations of a fundamental biodiversity conservation dilemma [73] between a biocentric paradigm of conservation focusing on maintaining natural processes in the ecosystems, and an anthropocentric paradigm [74]. The latter implies active management aiming to create and/or maintain a fixed composition of species and habitats desirable from the dominant stakeholders' viewpoint because of their socioeconomic, cultural, or ideological reasons [6].

Both the Polish and Belarusian conflicts involved both supporters and opponents of conservation, despite belonging to the same governmental system. In the Polish case, the entire disputed land is state-owned, and both the Białowieski NP and State Forest Holding are public institutions being supervised by Poland's Ministry for Environment. In the Belarusian case, the internal conflict of the years 2002–2008 broke out within the NP "Biełavieskaja Pušča" functioning as a diversified state-owned enterprise with a permanent staff of over 900, ranging from operating an own sawmill [64] to fundamental studies in conservation biology.

Both conflicts have been fought on the "aliens vs. locals" battleground, though in a case-specific way and with reverse outcomes. In the Polish case, the local population mostly supported the foresters' side of the dispute [28,30,66,70] whereas the proponents of extension of the passive protection regime have often been seen in the public discourse (see e.g., [66,70]) as "incomers" being countered to the autochthonous population in terms of their ethnicity, confession, and worldview.

When it comes to the Polish part of the Białowieża Forest massif's stewardship dilemmas, the voice of local communities is to a great extent treated in the public discourse as more legitimate, and thus superior to the opinions of experts and stakeholders from elsewhere. The latter statement, however, seems disputable as the issue of national park extension by definition belongs to the national-level decision-making domain [59,69].

Regarding the Belarusian part, the roles of locals vs. newcomers were inversed. Here, a group of locally residing staff (deputy director included) was dismissed in 2002–2008 by the newly appointed incomer director for their dissent against intensified logging of the bark beetle-infested forests including in the strict control area, and a new incomer workforce

was seized to do this job instead. Interestingly, those seasonal lumbers were popularly nicknamed "the Taliban" by many locals, thus addressing their lack of appreciation of the Białowieżą Forest massif's values being alien to many of them.

It is worth noticing that the local populations in these two different contexts were similar in their ethnicity, mother tongue, and religious confession. Thus, the share of ethnical Belarusians varies from 40% in Hajnówka powiat of Poland to 88% in the Pružany district of Belarus; they speak the same local dialect at home. Whilst in both the Polish and Belarusian parts of the Białowieża Forest massif people traditionally belong to the Orthodox (historically, mostly Greek-Catholic before 1839) church or to Neo-protestant denominations [75], vs. the proportion of the Polish/Roman Catholic population in the Białowieża Forest massif region is reversed.

Both disputes were so far resolved due to interventions by external actors rather than through local conflict mediation efforts; be it a decision of the European Court of Justice overruling logging in the Polish case, or a staffing solution of the presidential affairs office in the Belarusian case.

Unlike the Belarusian part of the Białowieża Forest, considerable literature addresses the negative attitude of the local population in the Polish part toward expanding the national park regime. Many local people in the Polish part expect that an enlarged protected area would impose limitations similar to those of the Białowieski NP strict reserve [31,76]. At the same time, some [32] claim that local consumption of forest provisioning ecosystem services such as timber, firewood, or forest fruits are declining with the distance from the Białowieża Forest massif. Hence, households located within the Białowieża Forest massif are more concerned than households located in its outskirts. Moreover, lower gains of the municipalities' budgets might demotivate local communities, because according to actual regulations, the municipal income tax rate per hectare of protected area is half as high compared to managed forest. As a consequence, reduced external investments are expected from an extension of the passive protection regime, which would entail a backlog in development, income decline, and growing unemployment [28]. Finally, intangible factors might also have an impact, such as ideological or religious beliefs, family historic experience, traditional knowledge [28,52,77], family or personal liaison with foresters [30], aesthetic preferences, and sentiments.

### 2.3. Survey Questionnaire and Interviews

We collected data using both quantitative and qualitative methods. First, a survey design using a questionnaire targeting owners of ecotourism businesses enabled us to collect quantitative data, which were subject to further econometric analysis. Second, to elicit the target groups' preferences using qualitative data, providing a deeper understanding of the factors shaping the ecotourism business owners' preferences, we also conducted in-depth semi-structured interviews with the same respondents. In this article, we report the quantitative data analysis, whereas the qualitative data presented here are limited to capturing quotations interpreting our qualitative results with particularly insightful sentences by the respondents.

The interviews were made throughout the transboundary Białowieża Forest massif area face-to-face in the respondents' homes between June and October of 2020 and with 24 booster phone interviews in the Belarusian part in late 2021. The sample consisted of owners of B&Bs, agro-tourism lodges, and hotels operating in the Białowieża Forest massif. When interviewing, we filled in paper questionnaire forms and simultaneously made use of a digital voice recorder, unless a respondent objected. Compared to the total number of ecotourism businesses in the Białowieża Forest massif and its immediate outskirts derived using multiple sources (e.g., official statistics, national parks' internal data, Google maps) we managed to cover over 50% of the active ecotourism businesses in the Belarusian and Polish parts of the Białowieża Forest massif and its immediate outskirts. This resulted in 107 respondents in Poland and 45 in Belarus. The difference in sample size reflects a far

more advanced development of the hospitality industry in the Polish segment than in the Belarusian part of the Białowieża Forest massif.

The development questionnaire and interview manual was an iterative process involving both experts and laypersons. In order to measure owners' attitudes toward conservation more accurately, we asked them the main attitudinal question using a five-point Likert scale (*2 = I strongly support, 1 = I somewhat support, 0 = I neither support nor not support, −1 = I somewhat do not support, −2 = I strongly do not support*) in the country-specific form. In the case of Belarus: *Is it enough to have a strict protection zone on about 40% of the national park area to preserve the nature of the Białowieża Forest massif, or should its whole area be a strict protection zone? Would you support such an idea?* In the case of Poland: *From the point of view of your activity, would it be good if the national park was extended to cover the whole area of the Białowieża Forest massif?* Because of small sample sizes, but covering >50% of the total population of nature-based tourism businesses, in both countries, we subsequently recoded the answers to the attitudinal question into a three-point Likert scale where the possible answers were: *1 = I support, 0 = I neither support nor not support* or *−1 = I do not support*. In addition to the attitudinal question, we asked a series of closed and open-ended questions aimed at better understanding the respondents' attitudes toward the Białowieża Forest massif's conservation. This approach provided a set of potential explanatory variables, which are discussed below.

Concerning *distance from the main villages Białowieża/Kamianiuki,* hosting headquarters of the national parks in Poland and Belarus, respectively, as a possible factor, we related to the studies indicating that the further a respondent resides from Białowieża Forest massif—the more positive the attitude toward conservation is [32], assuming the same regularity on the Belarusian side. To test if the size of the agro-tourism lodges impacts the owners' preferences we used the *number of beds* available at the farmstead as an explanatory variable. In addition, we used a set of dummy variables coding if a person *is local* (i.e., being born or living most of her/his life in the Białowieża Forest massif), if a person is *associated with the State Forest Holding* (a variable was used in the Polish model only) (i.e., if the closest family were or are working for the State Forest), and if running a B&B farmstead is the main source of the household income. Since, during pilot interviews, it turned out that owners specializing in hosting foreign visitors have a noticeably different attitude toward conservation, the variable "share of foreign visitors" was included in the estimated models.

To describe the ecotourism business owners' attitudes toward the Białowieża Forest conservation we estimated two country-specific ordered logit models. A detailed description of our econometric modeling approach is given in Appendix A.1 in the Supplementary material. The descriptive statistics of the variables are reported in Table 3. In the next section, we present and discuss the results.

**Table 3.** Descriptive statistics of the used variables.

| Variables | Country | | | | | | | |
|---|---|---|---|---|---|---|---|---|
| | Poland | | | | Belarus | | | |
| | **Mean** | **SD** | **Min** | **Max** | **Mean** | **SD** | **Min** | **Max** |
| Dependent variable: Attitude (3-point Likert scale) | −0.103 | 0.910 | −1 | 1 | 0.422 | 0.839 | −1 | 1 |
| Independent variables: | | | | | | | | |
| Distance from the main villages | 18.56 | 19.30 | 1 | 64 | 19.71 | 13.69 | 1 | 40 |
| Number of beds | 14.37 | 12.54 | 2 | 60 | 16.87 | 10.54 | 4 | 41 |
| Main source of income | 0.28 | 0.45 | 0 | 1 | 0.51 | 0.51 | 0 | 1 |
| Share of foreign visitors | 15.78 | 18.36 | 0 | 80 | 32.44 | 23.47 | 0 | 80 |
| Associated with State Forest | 0.27 | 0.45 | 0 | 1 | — | — | — | — |
| Being local | 0.69 | 0.46 | 0 | 1 | 0.47 | 0.50 | 0 | 1 |
| Sample size | 107 | | | | 45 | | | |

## 3. Results

As a direct interpretation of quantitative modeling coefficients' estimates in the ordered logit models is difficult, to interpret the results, we focused on the coefficients' signs and statistical significance (Table 4). All six coefficients in the Polish case as well as two coefficients out of five in the Belarusian case were statistically significant at least at a 95% confidence level. Given that one coefficient in the Belarusian case (i.e., *being local*) was not significant, whereas, for two more coefficients (i.e., *distance from the main villages* and *number of beds*), their confidence intervals included both negative and positive values, the overall significance level in the case of Belarus is lower than in Poland. At the same time, despite a relatively small sample size, all estimates for Poland except for the *number of beds* turned out to be significant at the 99% confidence level.

**Table 4.** Ordered Logit model results for Poland and Belarus. The dependent variable reflects attitudes toward increased area protection in the Białowieża Forest. In both countries, the possible levels of the dependent variables were: *I support* (coded as 1), *neither support nor not support* (coded as 0), *I don't support* (coded as −1).

| Variables | Poland | | Belarus | |
|---|---|---|---|---|
| | Coefficient | (Std. Error) | Coefficient | (Std. Error) |
| Distance from the main villages | 0.038 *** | (0.012) | −0.071 * | (0.042) |
| Number of beds | −0.056 ** | (0.022) | 0.068 * | (0.058) |
| Main source of income | 1.337 *** | (0.514) | 1.925 ** | (0.909) |
| Share of foreign visitors | 0.058 *** | (0.016) | 0.055 ** | (0.024) |
| Associated with State Forest | −1.807 *** | (0.590) | − | − |
| Being local | −1.600 *** | (0.484) | 0.605 | (1.044) |
| Model characteristics | | | | |
| Log−likelihood | −81.161 | | −27.370 | |
| McFadden's pseudo $R^2$ | 0.259 | | 0.240 | |
| Sample size | 107 | | 45 | |

***, **, * significance at 1%, 5%, 10% level.

The signs of the statistically significant parameters were consistent across the two country-specific models. Thus, in both Poland and Belarus, a higher *share of foreign visitors* and offering ecotourism services as a *main source* of households' income positively co-varied with the higher probability of their owners' support for increased area protection. Other tendencies were specific to Poland, where the facts of *being local* and being *associated with the State Forest Holding*, as well as a greater *number of beds* available in the farmstead, decreased the probability of its owner's support for the extension of the Białowieski NP. At the same time, none of the explanatory variables was found to significantly reduce the probability of support toward the extension of the strict control zone in the Belarusian case. To better understand the impact of the independent variable, in addition to reporting ordered logit models' parameter values (Table 4), we calculated their marginal effects (Table 5). There were two types of explanatory variables used in the ordered logit model; continuous (i.e., *distance, number of beds*, and *share of foreign visitors*) and discrete (*main source, being associated with the State Forest Holding* and *being local*). In the case of the continuous variables, the marginal effect represents the change in the probability for each level of attitude if the explanatory variable changes by 1 unit. In the case of a dummy variable, the marginal effect represents the change in the probability resulting from the change in the explanatory variable from 0 to 1. The marginal effects with corresponding 95% confidence intervals for having a positive attitude toward increasing protected areas are presented in Figure 3.

We see that in the group of dummy variables taking binary values of either unity or zero, the largest impact on attitude toward the conservation of the Białowieża Forest on the Polish side was *associated with State Forest.* The probability of having a negative attitude increases by 0.28 if an owner liaises with the State Forest (keeping all other explanatory variables at their means). Slightly smaller in absolute terms but also very substantial was

the effect due to discrete change (from zero to unity) in *being local* and in *main source* of income. One unit change in these variables will increase the probability that a person will have a negative attitude by 0.25 and −0.21, respectively. Concerning the continuous variables, a one-unit change in *share of foreign visitors* and *number of beds* has a similar impact on attitudes in terms of strength. However, pointing in the opposite direction, an increase in *share of foreign visitors* by one unit (by 1% in this case) will decrease the probability of having a negative attitude by 0.01, whereas an increase in *number of beds* by one will increase the probability of having negative attitude by 0.01. An impact in terms of the magnitude of the *distance from the main villages* continuous variable is that an increase of 1 km will decrease the probability of having a negative attitude by 0.006. Concerning the Belarussian side, all the highly significant estimates of marginal effects increase the probability of a positive attitude toward expanding spatial protection. Thus, offering ecotourism services as a household's main source of income increases the probability of its owner's positive attitude by 0.25, whilst an increase in the *share of foreign visitors* by 1% increases the probability thereof by 0.007.

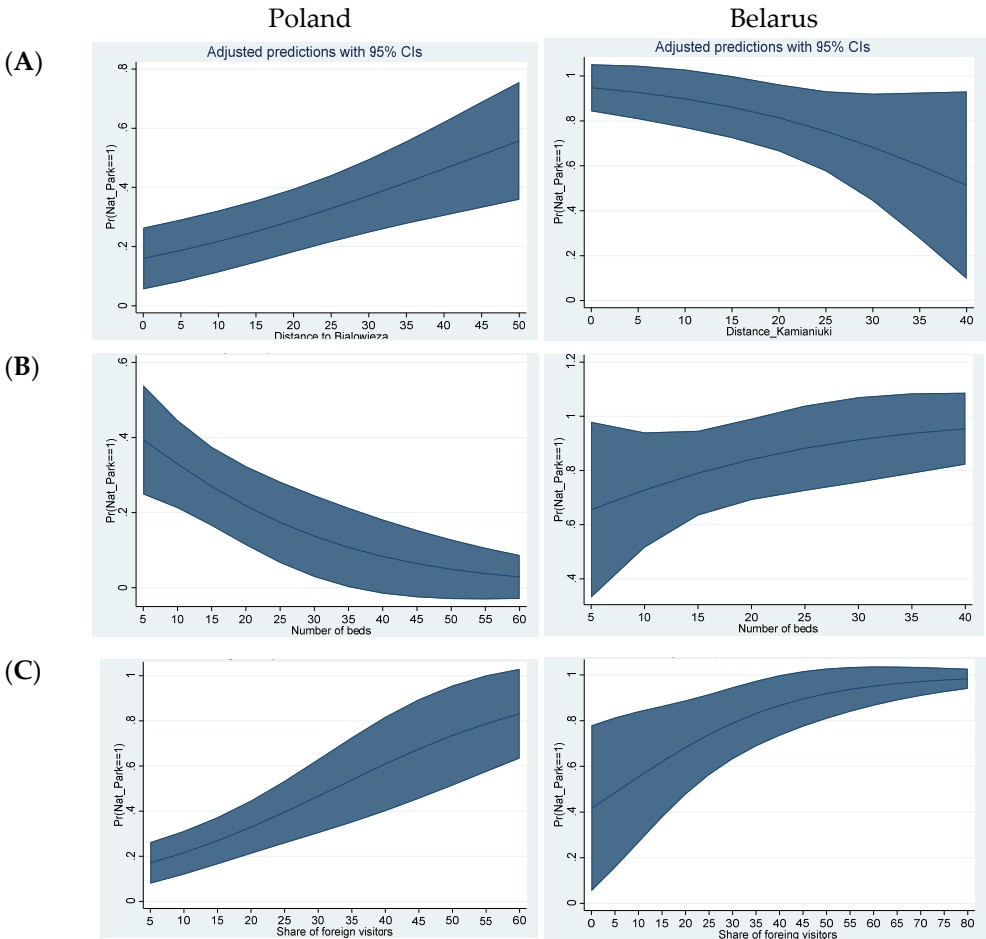

**Figure 3.** *Cont.*

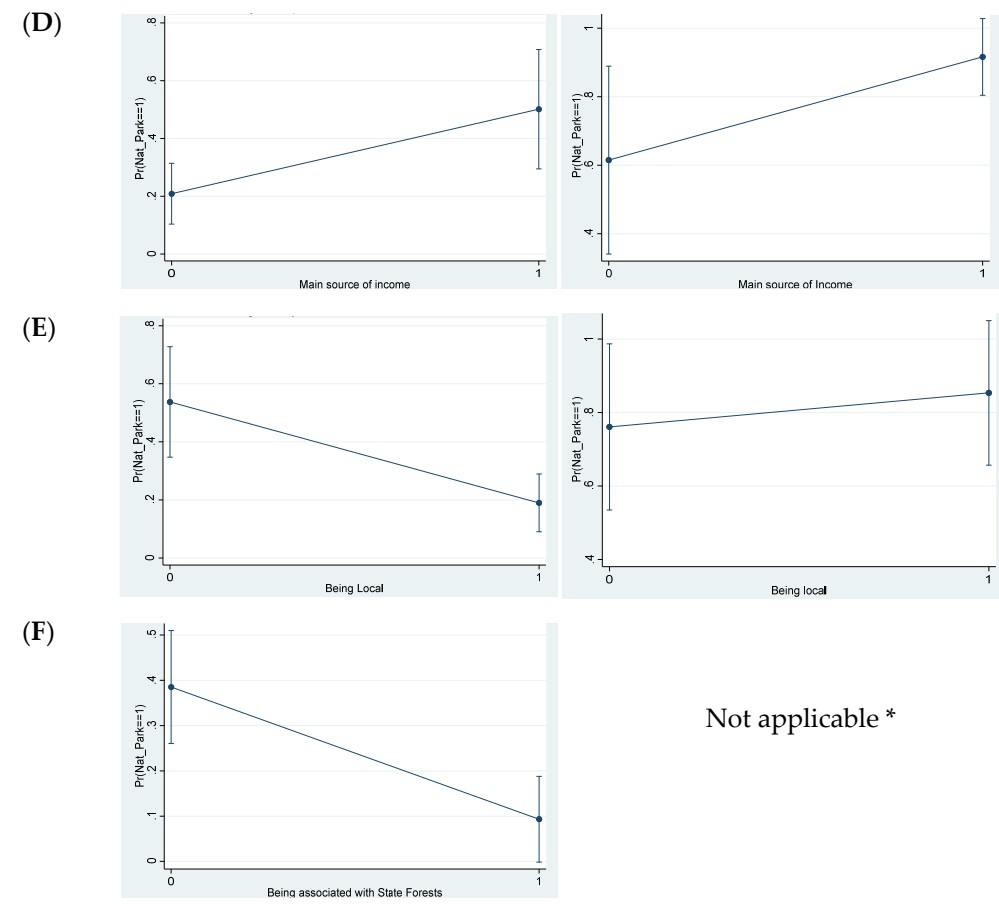

**Figure 3.** Marginal effects of the explanatory variables (depicted on the horizontal axis) for positive attitude (answer: *I support*, depicted on the vertical axes) toward extending protection area in the Białowieża Forest massif among Polish and Belarusian owners of ecotourism businesses. Explanatory variables: (**A**) Distance to the main villages, (**B**) Number of beds, (**C**) Main source of income, (**D**) Share of foreign visitors, (**E**) Being local, (**F**) Being associated with State Forests in Poland (* was not used in the model for Belarus).

**Table 5.** Marginal effects for different variables affecting attitudes toward increasing conservation level in the Białowieża Forest.

| | Attitude | | | | | |
|---|---|---|---|---|---|---|
| | **Poland** | | | **Belarus** | | |
| **Independent Variables** | **Negative Margin. Eff. (*p*-Value)** | **Neutral Margin. Eff. (*p*-Value)** | **Positive Margin. Eff. (*p*-value)** | **Negative Margin. Eff. (*p*-Value)** | **Neutral Margin. Eff. (*p*-Value)** | **Positive Margin. Eff. (*p*-Value)** |
| Distance from the main villages | −0.006 *** (0.002) | 0.000 (0.000) | 0.006 *** (0.002) | −0.006 * (0.003) | 0.003 (0.002) | 0.009* (0.005) |
| Number of beds | 0.009 *** (0.003) | −0.000 (0.000) | −0.008 *** (0.003) | −0.006 (0.005) | −0.002 (0.003) | 0.009 * (0.007) |
| Main source of income | −0.207 *** (0.073) | 0.005 (0.012) | 0.202 *** (0.071) | −0.175 ** (0.075) | −0.078 (0.046) | 0.253 ** (0.102) |
| Share of foreign visitors | −0.009 *** (0.002) | 0.000 (0.001) | 0.009 *** (0.002) | −0.005 *** (0.002) | −0.002 * (0.001) | 0.007 *** (0.002) |
| Being associated with State Forests | 0.279 *** (0.078) | −0.007 (0.015) | −0.273 *** (0.084) | — | — | — |
| Being local | 0.247 *** (0.063) | −0.006 (0.013) | −0.241 *** (0.064) | −0.055 (0.097) | −0.025 (0.041) | 0.080 (0.136) |

***, **, * significance at 1%, 5%, 10% level.

## 4. Discussion

### 4.1. Opposite Attitudes Underpinned by Generic Dependencies

Attitudes of the ecotourism business owners toward strict protection of a larger proportion of the Białowieża Forest massif showed opposite patterns across the Polish-Belarusian state border. Whilst the sample of Polish respondents was (on average) negative toward the extension of the Białowieski NP over the entire Białowieża Forest massif's western segment, their Belarusian counterparts (on average) were supportive of an extension of the strict control zone regime over the entire Belarusian part of the Białowieża Forest massif. These patterns are consistent with empirical literature presenting mixed evidence regarding the overall environmental attitudes of hospitality business owners and/or managers (see e.g., [36–38,42–44,78,79]). The novelty of our results lies in eliciting their preferences toward extended biodiversity protection and stewardship, which goes beyond internal hotel management but has a direct impact on the prosperity of their business. Our results suggest that the ecotourism business owners of the Polish and Belarusian parts of the transboundary forest massif view this impact differently.

At the same time, significant dependencies between explanatory variables and the attitude toward increased area protection show no contradictions between Poland and Belarus. Thus, providing hospitality services to Białowieża Forest massif visitors as the B&B businesses' main source of income was positively correlated with their propensity to support strengthening the nature protection regime in the Białowieża Forest massif in both countries. A higher material dependency on the visitors' inflow to the Białowieża Forest massif motivates the agro-tourism lodge owners regardless of country of residence to be on average more appreciative of the Białowieża Forest massif's higher level of naturalness, and therefore be supportive toward a greater level of their protection.

The same tendency applied in both countries to the share of foreign visitors where a higher percentage of visitors from abroad increases the probability of respondents' support toward increased passive protection. The underpinning reason might be a more pronounced wildlife-oriented profile of foreign visitors compared to domestic visitors. Thus, 80% of foreign tourists visiting the Białowieża Forest massif between March and mid-June want to watch birds, compared to about 20–30% of Polish visitors [80]. Furthermore, hosting international birdwatchers appears to be more profitable for agro-tourism farmstead owners, because birdwatchers tend to stay longer and spend more money onsite compared to an average visitor [80]. This category of visitors is sharply oriented toward protected areas (e.g., [81–86]), whereas the preferences of hospitality business owners appear to be a consequence of the preferences of their target clients [87].

The rest of the significant tendencies were more pronounced for Poland than for Belarus. The Białowieża village is the biggest settlement in the Białowieża Forest massif, hosts the headquarters of the Białowieski NP, and is the locus of the highest concentration of hospitality businesses in the Polish part of the Białowieża Forest massif. A bigger distance from Białowieża to respondents' business reduced the probability of reluctance toward greater spatial protection of the Białowieża Forest massif. This is consistent with findings of, e.g., [32] stating that this tendency might be explained by a lower financial dependence of the respondents residing outside the Białowieża Forest massif on the accessibility of the forests' ecosystem services (including the forests' accessibility for their clients); hence their lower sensitivity toward expected access restrictions. Respondents residing in the Białowieża village, where the intensity of public discussion and conflict over the hypothetical enlargement of the national park is higher, can make a stronger impact on their preferences. Moreover, some of the B&B owners located within the Białowieża Forest massif might consider the Białowieski NP a rival to their own hospitality business as it operates its own guest rooms and thus competes with the other ecotourism businesses' operators on the market. Unlike those located in Białowieża, the owners of agro-tourism lodges from the outskirts of the Białowieża Forest massif can apply a more diversified strategy, which does not exclusively rely on the Białowieża Forest massif itself as a tourist destination. A Polish respondent commented: *"Our guests are not exclusively oriented towards*

*the Białowieża Forest massif, as they may spend just one day in Białowieża, and also go to visit other destinations in the region of stay*". In principle, this tendency is also relevant for the Belarusian part, although observed there at a lower level of significance. Another possible explanation is that according to [88], tourism-oriented development may increase locals' living costs, affecting minor B&B farmstead owners and thus shifting their preferences away from local "overdevelopment".

A reverse tendency might apply to the total number of beds in guest rooms in the respondent's possession, a parameter which in accordance with the model estimated for Poland negatively affects the probability of the respondent's support of increased strict area protection. The underlying reason may be associated with the tendency of more sustainable and wildlife-oriented visitors to travel individually or in small groups. Unlike them, participants of organized game tours or foresters' corporate meetings tend to require facilities with more beds to host their stay in the Białowieża Forest massif. This tendency was not observed with the Belarusian respondents.

Two remaining variables, which affect negatively the probability of supporting the increased area protection over the entire Białowieża Forest massif's Polish part, were local origin and connection with the State Forest Holding. This makes sense because both people of local origin, and those liaised (e.g., through their professional and/or family/personal ties) with the State Forest Holding and/or with individual foresters, are widely reported in the literature as the social groups largely opposed to increased area protection (e.g., [31,66,70]). For Belarus, because the appropriate coefficient did not statistically differ from zero, the fact of respondents' local origin appeared neutral to extending the strict protection zone.

### 4.2. Why, Unlike in Belarus, Increased Area Protection Is Not Popular in Poland

Our qualitative results suggest that unlike in the Polish part of the Białowieża Forest massif, in its Belarusian part, the current state of spatial nature protection represents a socially supported equilibrium [26,27]. Indeed, the historical core of the Belarusian part of the Białowieża Forest massif is largely covered with the recently expanded strict control regime precluding it from any considerable human-induced interference. Non-recreational impact over the remaining three national park zones (Table 2) is restricted. As a result, among less than a quarter of the Belarusian respondents who appeared reluctant to further extension of the strict control zone to cover the entire area of the NP "Biełavieskaja Pušča", some still revealed a pro-conservationist stand. Correspondingly, over $\frac{3}{4}$ of the respondents, irrespective of their characteristics under scrutiny, appeared supportive or neutral toward the further extension of area protection mostly because of their allegedly insufficient expertise. At the same time, being asked about the current and desired cooperation with the national park administration, the majority of Belarusian respondents merely appreciated the lack of interference by the park administration. This implies that respondents in the Belarusian part of the Białowieża Forest massif distinguish between the national park as a form of spatial protection, and the national park administration as an influential top-down stakeholder. Whilst the former is overwhelmingly appreciated, the latter seems to have a somewhat mixed profile. A minority of the respondents revealed their liaison with the national park administration.

On the contrary, according to Polish respondents' statements regarding the prospective extension of the national park, the conflict between use and conservation remains. Thus, respondents stating unambiguously negative attitudes (35%) claim that salvage logging following bark beetle disturbances should be increased to fight the bark beetle throughout the Białowieża Forest massif including in its nature reserves. The respondents who state a less negative attitude (12%) were still strongly in favor of increasing logging, however, with the exception of the nature reserves.

The attitudes of respondents stating their neutrality with respect to protected area extension (13%) ranged from complete indifference to a preference for some "golden balance" between strict area protection and forestry. Many respondents in this group

abstained from stating any clear position, as they associated it with extreme solutions. Instead, they tended to partly agree with the foresters' and partly with the conservationists' argument, whereas some stressed their neutrality in this dispute, claiming a lack of expertise. Notably, the two biggest hotel operators in the Białowieża village refused to participate in the survey, presumably being cautious of the risk of being perceived as partisans amidst the local conflict. A frequently repeated statement in this group was that the dispute over the Białowieża Forest massif is really a dispute among stakeholder groups about money: "ecologists" are claimed to be paid for their activities, and the State Forest also makes money from the forest.

Respondents presenting a rather positive attitude to increased area protection (12%) believe that the current level of protection is insufficient to maintain the Białowieża Forest legacies of naturalness but does not necessarily cover the entire massif. Some respondents (especially, in and near the Białowieża village) seemed cautious to take an unambiguously positive stand, in striking contrast to the respondents reluctant toward protection, who did not hesitate to openly express their opinions. Furthermore, some positively inclined respondents expressed skepticism toward the Białowieski NP as an institution. Finally, the respondents with an unequivocally positive stand (28%) clearly stated their support to increase the protected area and the level of nature protection in the Białowieża Forest massif.

It should be emphasized that the people who declared a negative attitude to extended protection, predominantly still declare their care about the nature values of entire the Białowieża Forest massif. However, they consider its current (as well as more pro-conservation) stewardship model detrimental because according to them, the Białowieża Forest massif is presented as dying, not being able to cope with the bark beetle without human intervention. Some people (mainly the elderly) complained that they would never see again the nice "primeval" Białowieża Forest massif from the times of their youth. The idea of extending protection is thus rejected because of its impact on the aesthetic attractiveness of the Białowieża Forest massif in their perception (*"it would be nice to have more noble broad-leaved deciduous trees planted in the Białowieża Forest"*). The sight of deadwood evokes negative reactions, especially toward large amounts of felled trees that have been piled up in some places. Explaining their negative reactions, they most often mentioned safety, economic, and aesthetic reasons (*"dead stands are dangerous for visitors"*, *"valuable material is wasted in vain"*, *"an ugly sight of deadwood repels tourists"*). As the latter aesthetical impressions are mostly derived from the Białowieża Forest's massif managed part, the findings of [89] seem relevant because the psycho-relaxing properties of a forest landscape with deadwood are better in a naturally dynamic forest than in a managed forest.

Finally, the high impact of the State Forest Holding on shaping the Polish respondents' preferences is traceable from the observed behavior of respondents because their argument to oppose protection extension almost literally follows the narratives of a State-Forest-issued popular leaflet [90]. For instance, a frequently repeated narrative is that having failed to proactively fell and remove small groups of bark beetle-infested trees in time, led to infestation on a mass scale and now the forest is doomed. The in-depth interviews indicate that the narrative on bark beetle and spruce stands' mass die-off is influential also for some of the supporters of national park extension, as some of them argued that action should be taken, even in the strict protection area, if it stops degradation.

Besides the essence of the conflict, a striking difference in attitudes among Polish respondents was observed. Whilst the foresters were largely respected because *"the Białowieża Forest is so beautiful and famous as a result of devoted effort of generations of foresters who have created and maintained its value"*, the national park extension supporters (being locally defined as *"(false-) ecologists"*, in Polish: *"(pseudo-) ekolodzy"*—see, e.g., Figure 4A) were presented in different shades of negative light. This ranged from *"poor freaky youngsters lacking a single small coin to spare who were sleeping under the open sky nearby my barn"* to *"highly paid agents of foreign lobbyists who lived in my farmstead drinking extremely luxury vodkas—such that you have never tried"*.

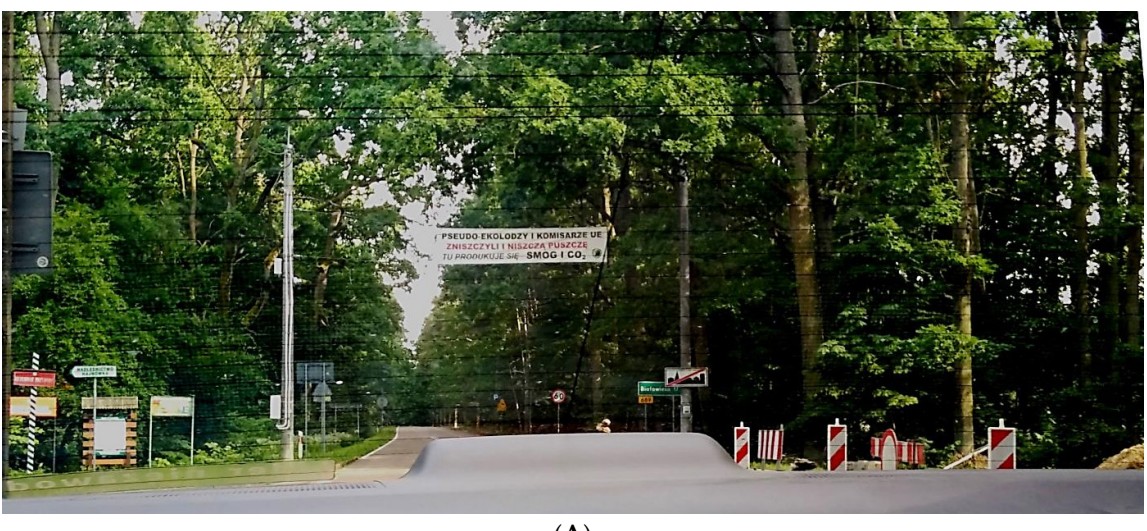

(**A**)

**Figure 4.** *Cont.*

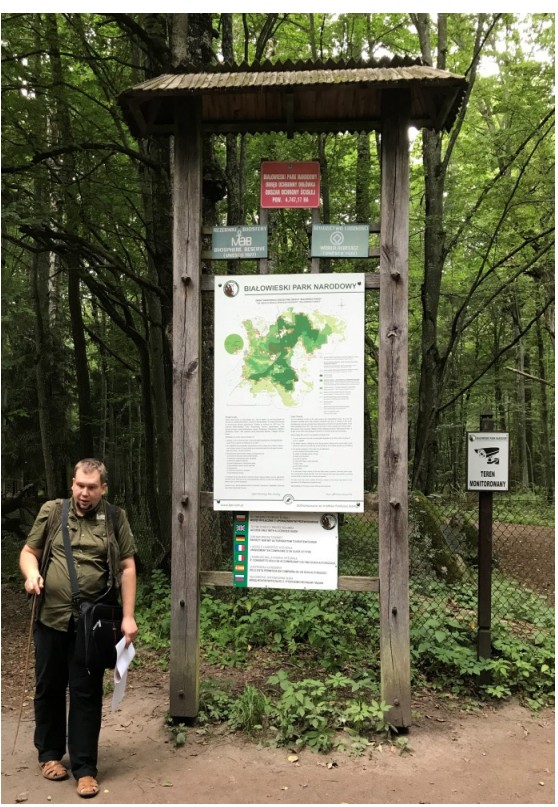

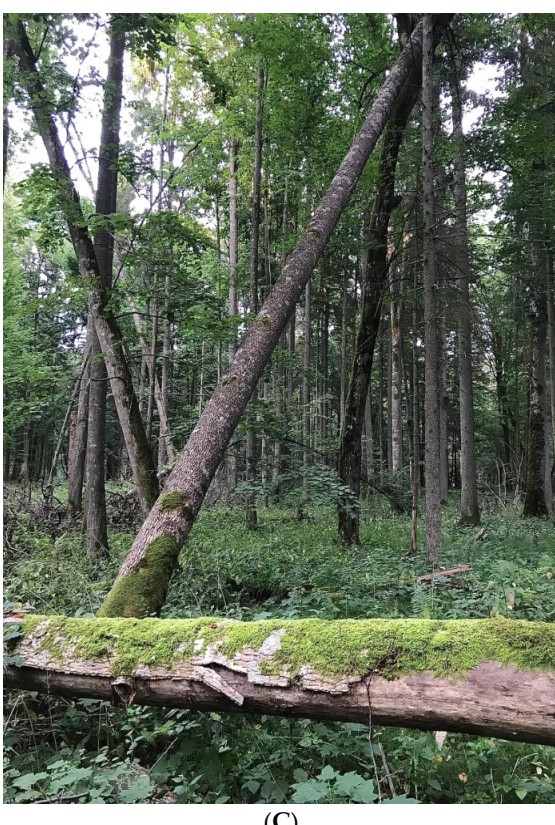

(**B**)                                                                 (**C**)

**Figure 4.** Manifestations of the forest landscape stewardship conflict in the Białowieża Forest massif (**A**): the banner over the R689 Hajnówka—Białowieża motorway stating *"False-Ecologists and EU Commissioners have devastated and still devastate the Białowieża Forest. Smog and $CO_2$ are being produced here"*—photo by Sviataslau Valasiuk taken in June 2020. Protected areas (**B**) that form functional networks of habitats with high levels of naturalness (**C**) are crucial for supporting transitions toward multifunctional landscapes (Photos by Per Angelstam).

The presence of highly influential stakeholders opposing increased area protection, namely the Polish State Forest Holding and foresters as a distinctly professional and social corporation, is what clearly differentiates the Polish and Belarusian parts of the

Białowieża Forest massif. Historically, since the establishment of independent Poland, forestry servicemen have been personalizing the raison d'état in the Białowieża Forest massif area being largely inhabited by ethnic and confessional minorities. They traditionally enjoyed a high social profile in terms of provisioning ecosystem services' stewardship (e.g., issuing permissions on timber and firewood, game, non-forest products), as well as being the most prestigious employer and provider of public commodities (e.g., road construction, forestry technical education, and social mobility) to the local population. Simultaneously, the State Forest Holding maintains good political connections locally [30]. Unlike the State Forest Holding, the Białowieski NP is considered a local lower-profile institution with lower salaries compared to State Forest. The restrictions imposed by the Białowieski NP are seen by the local community as hardly consistent with the "traditional knowledge", i.e., the knowledge derived from the locally prevalent industrial forestry model originating from 19th century Germany [91].

The findings of our quantitative analysis match this situation well. Indeed, the more a respondent liaises with the State Forest domain via professional and/or personal/family connections, and/or the more the respondent's family was historically dependent on foresters' benevolence, the more likely they are prone to reproduce the State Forest narratives regarding the harmfulness of extending the national park. The reasons are being a person born inside the Białowieża Forest massif, and hospitality businesses being oriented toward the foresters' and/or hunters' market segment. Therefore, an apparent tendency of the Polish respondents to "vote" against their own economic interest is rational because of the absolute dominance of the State Forest Holding as a local stakeholder. On the contrary, the more a person is dependent on the inflow of tourists to the Białowieża Forest's different zones of conservation and use, and the weaker the current or past dependence on the Białowieża Forest's provisioning ecosystem service is under State Forest control, the less opposed they are to increased area protection. This applies especially to highly wildlife-oriented international visitors traveling individually or in small groups.

### 4.3. Zoning and Governance at Landscape and Regional Scales

No single forest management regime is able to conserve biodiversity, maximize wood yield, store carbon, and provide multiple ecosystem services as a base for multiple value chains. For example, both uneven-aged and mature even-aged forests contribute to the maintenance of biodiversity; however, sufficient amounts of natural forests forming functional habitat networks are also needed to ensure the future of forest-dependent species [92]. Given that a broader set of biodiversity aspects need to be protected, the best overall biodiversity impacts for a variety of species at the landscape level can be achieved by ensuring that there is a mosaic of different forest management regimes within landscapes and regions [93]. Given the long history of loss in Europe of natural forest remnants such as the Białowieża Forest massif, their conservation is of paramount importance. Viewing forest landscapes as social-ecological systems is crucial (Figure 4B,C).

Key barriers toward the evolution of multifunctional landscapes include solid long traditions of even-aged forest management focusing on maximum yield [91,94], strong professional guilds exhibiting inertia to change [6,95], and difficulties in introducing spatial planning that segregates different functions at relevant spatial scales [96]. This and previous studies about Białowieża Forest stakeholders' and actors' different preferred values, e.g., [28,30–32,97] illustrate the challenges of conserving the few last European remnants of high conservation value forests, including their species, habitat networks, and ecological processes. Many other studies provide evidence of similar and other challenges rooted in the conflict between nature and society from Central and Eastern Europe, e.g., [7–10] and in a broader geographical context, e.g., [98]. Clearly, there is a need for very large conservation areas, especially because both natural forest disturbance regimes and interactions between large herbivore and carnivore populations are target functions [6]. With the state being the dominating forest owner in both the Polish and Belarusian parts of the Białowieża Forest

massif, in principle, the necessary landscape and even regional perspective should be easier to implement than in settings with many different small landowners.

Indeed there are zoning systems in Poland and Belarus, which could bring together different stakeholder groups, mitigate societal tensions, and thus facilitate forest landscape stewardship. For instance, the Polish Promotional Forest Complex (PFC) concept was introduced in the 1990s to implement policies about sustainable forest management in 19 pilot areas [97]. The first PFC was established in the Białowieża Forest [28] in 1994. It was chosen as an alternative to the non-interventionist approach of expanding the Białowieża National Park based on active forest management. However, results of [97] indicate that PFCs became neither pioneers in implementing pro-ecological forestry nor exemplary models for other Polish forests. This illustrates that sustainable development, as a process toward the goal of sustainability in countries in transition, requires a change in attitudes among stakeholders that reflect both SFM policy, transitions from government to governance, and appreciation of new value chains built on high conservation forest values. In the Belarusian part, an approach to zoning reflects the character of a national park as a diversified enterprise that needs to combine the statutory goal of biodiversity conservation with commercially oriented activities based on material forest resources use outside the strict control zone. Whilst the surveyed Belarusian stakeholders have stated their appreciation of the former, they are still somewhat cautious toward the latter as they seem to treat the national park administration as a potential rival dominating its market power. The multifunctionality of landscapes implemented through a zoning system increases their ability to obtain market niches and sustainably develop different value chains. This need for transitions toward multifunctional forest landscapes applies also to other remnant biodiversity hotspots in Europe, such as intact forest landscapes in Fennoscandia [1] and northwest Russia [99], and large forest massifs in Eastern Europe [100].

Supporting the vision of multifunctional forest landscapes, we argue that the zoning approach applied in the Białowieża Forest massif is a good start, but that the proportion and spatial extent focusing on sustaining all aspects of biodiversity conservation should be the focus. However, given the large area required to sustain ecological processes, even the Białowieża Forest massif is small. Therefore, as suggested by [42], we advocate that this remnant island of the once widespread temperate lowland forest needs to be subject to a regional planning perspective as well as nature restoration [101]. This should include the archipelago of remnant forest patches with different management and conservation legacies along the entire EU's eastern border in both Poland and Belarus (Figure 1). However, there are barriers ranging from geopolitical developments hampering transboundary collaboration to the tall fence built along the Polish-Belarusian border, which does not allow the movement of large herbivores and carnivores.

## 5. Conclusions

Comparing the results of the questionnaire survey administered to the Polish and Belarusian subsets of the transboundary Białowieża Forest massif's local agro-tourism farmstead owners, and interviews with providers of tourism services, we show that the country-specific context makes some Polish ecotourism business owners oppose national park extension. In contrast, Belarusian ecotourism business owners behave rationally in their own interest to support park extension. This pattern is linked to the fact that, unlike in the Belarusian part, the Polish part of the Białowieża Forest massif extension of the national park lacks a high-profile local institutional "ambassador" supporting conservation. The dominance of the NP "Biełavieskaja Pušča" in the Białowieża Forest massif's Belarusian segment clearly means that the local preferences are in favor of greater support of increased area protection. On the contrary, the Polish Białowieski NP in its current state has a rather low profile as a stakeholder amidst the debate on Białowieża Forest massif stewardship. As a result, even local supporters of increased forest protection amongst ecotourism business owners nevertheless demonstrate pronounced skepticism toward it. Moreover, State-Forest-inspired narratives condemning national park expansion are disseminated. Thus, the Polish

State Forest Holding has retained an influential position due to its historical role, its strong political connections, and efficient channels of disseminating its narratives shaping the local so-called "traditional knowledge" reflecting a focus on the need to manage forests.

Our results suggest that economic incentives stemming from nature-based tourism do not outbalance the State Forest influence even amongst agro-tourism farmstead owners, whose businesses depend on the Białowieża Forest massif as a unique natural resource for their business activities. Commitment to a conservationist standpoint is feasible only if the business owner is: (1) highly dependent on the inflow of tourists; (2) sharply focused on providing hospitality services to qualified international naturalists/nature-based tourists and birdwatchers traveling individually or in small groups; and (3) enjoys a greater degree of freedom from the dominating State Forest narrative. This is more likely with weaker social connections with the State Forest Holding and the local community or through businesses' geographical location in the outskirts rather than inside the Białowieża Forest massif. Given the corporate strength and a clear dominance of the State Forest Holding amidst the stewardship dispute in the Białowieża Forest massif, increased area protection is unlikely in Poland, unless the principle of local vetoing is abolished from the legislation. A key challenge is to bridge barriers maintained by different stakeholder perspectives on the transition toward multifunctional landscapes. A century of clashes between traditions of forest management focusing on maximum yield of wood, vs. sustaining resilient forests as complex ecosystems also forming the base for multiple value chains, need to transition to evidence-based collaborative learning and regional planning.

**Supplementary Materials:** The following supporting information can be downloaded at: https://www.mdpi.com/article/10.3390/land12061150/s1, Figure S1. Map of the Białowieża Forest massif with the borders of different denominations: Biosphere Reserve, World Heritage Site, and zones for conservation and use [60]; Appendix A.1: Econometric Framework.

**Author Contributions:** Conceptualization, M.G., S.V. and P.A.; methodology, M.G. and S.V.; software, M.G.; validation, M.G., S.V. and M.Y.; formal analysis, M.G., M.K. and J.M.; investigation, S.V. and M.Y.; resources, M.G., S.V. and M.Y.; data curation, M.G.; writing—original draft preparation, S.V., M.K. and J.M.; writing—review and editing, P.A., M.G. and S.V.; visualization, M.G. and P.A.; supervision, M.G.; project administration, S.V.; funding acquisition, M.G. and P.A. All authors have read and agreed to the published version of the manuscript.

**Funding:** The study has been carried out under an agreement between the University of Warsaw and the Swedish University of Agricultural Sciences (SLU), which was funded by the Swedish Research Council FORMAS (grant number 2017:1342 with Per Angelstam as project leader). The research conducted by Marek Giergiczny was funded by the National Science Centre (NCN), Poland. Grant number: 2021/43/B/HS4/03371.

**Institutional Review Board Statement:** Not applicable.

**Informed Consent Statement:** Informed consent was obtained from respondents, all of whom were involved anonymously in the study. The text has been stripped of any personal data.

**Data Availability Statement:** The data presented in this study are available on request from the corresponding author.

**Acknowledgments:** The authors are grateful to Vasil Arnolbik for his kind assistance. Thanks to Michael Manton for making maps, Jakub Bubnicki for comments, and Bogdan Jaroszewicz for providing insights about landscape zoning in the transboundary Białowieża Forest massif. We are also grateful to the anonymous reviewers.

**Conflicts of Interest:** The authors declare no conflict of interest, and the funders had no role in the design of the study; in the collection, analyses, or interpretation of data; in the writing of the manuscript; or in the decision to publish the results.

## Appendix A

*Appendix A.1. Econometric Framework*

In order to understand the owners' attitudes toward the Białowieża Forest conservation country-specific ordered logit models with dependent variable $y_i$ taking three levels and a set of $x_k$ explanatory variables were estimated. The estimated ordered logit model takes the following form:

$$y_i^* = \beta' x_i + \varepsilon_i, \text{ for } i = 1, \ldots, n,$$

where the continuous latent dependent variable $y_i^*$, is described in discrete form through an ordinal logit model:

$$\begin{cases} y_i = 0 \text{ if } -\infty < y_i^* \leq \mu_0 \\ y_i = 1 \text{ if } \mu_0 < y_i^* \leq \mu_1 \\ y_i = 2 \text{ if } \mu_1 < y_i^* \leq \mu_2 \\ \quad \cdots \\ y_i = J \text{ if } \mu_{J-1} < y_i^* \leq \infty \end{cases}$$

and where $\beta$ is a set of parameters estimated using maximum likelihood, with the assumption of $\varepsilon_i$ being a continuous random disturbance with cumulative distribution function (Greene, 2009) and the conditional variance of $\varepsilon$ being constantly equal to $\text{Var}(\varepsilon|x) = \pi^2/3$ (Long, 1997). The used distribution of errors $\varepsilon_i$ is a logistic, given by a function:

$$F(\varepsilon) = \frac{\exp(\varepsilon)}{1 + \exp(\varepsilon)}$$

An important assumption underlying the ordered logit model is a parallel line (known also as proportional odds) assumption, which means that the slope coefficients in the model are the same across all response categories (hence lines of the same slope are parallel). If the assumption holds, which is our case (i.e., $\beta_j$ are equal for all j-th levels $\beta = \beta_j$), then one equation for all levels of the response variable can be estimated.

As the parameters of an ordered logit model are hard to interpret in addition to presenting ordered model regression results, we calculated marginal effects. Marginal effects show the change in probability when the predictor variable increases by one unit. For a continuous variable, this represents the instantaneous change given that the "unit" may be very small. For binary variables, the change is from 0 to 1. Following (Green, 2009), a marginal effect of $x_k$ variable is calculated by an equation:

$$\frac{\partial \text{Prob}(y = j|x)}{\partial x_k} = \left[ f\left(\mu_{j-1} - \beta'x\right) - f\left(\mu_j - \beta'x\right) \right] \beta_{x_k}$$

All models were estimated in R.

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
