# Peer review of "Attitudes toward Conservation of the Transboundary Białowieża Forest among Ecotourism Businesses in Poland and Belarus"

_land, doi:10.3390/land12061150_

Round 1

Reviewer 1 Report

This manuscript examines attitudes of ecotourism businesses about potential extension of forest protection in the Bialowieza Forest in in Poland and Belarus. The manuscript has substantial potential to make an academic contribution but it also has weaknesses that need to be addressed.

First, the methodology section does not provide sufficient information on how the survey was administered and how the participants were recruited. Can the survey sample be characterised in some way, to provide background for interpreting results? The same is the issue with qualitative interviews. With whom they were conducted, how participants were recruited and how many they were?

The results section and the discussion do not really report any results based on qualitative interviews. Did the interviews yield anything distinct from the survey or just help interpret the survey results? In either case, it is necessary to note this explicitly.

Third, the results section overwhelmingly focuses on attitudes in Poland and those in Belarus are left with less attention. The results regarding Belarus do need elaboration and extension and more interpretation to achieve better balance in the focus. The results are such are interesting and valuable.

Finally, more effort should be made to reflect on the contribution of the article to wider peer literature. What is reported here that addresses gaps or weaknesses in the existing literature, how does this manuscript take the conversation forward? The academic contribution should also be communicated in the abstract.

Reviewer 2 Report

REVIEW the Article: "Attitudes towards conservation of the transboundary Białowieża Forest among ecotourism businesses in Poland and Belarus"

<Overall>

∙        Traditional forestry and forest-based tourism often lead to conflicts, and previous studies in European countries, including Germany, have been conducted on forest recreation and tourism for private forest owners.

∙        This study is highly significant, but understanding stakeholders' intentions through quantitative analysis alone is difficult. It is impressive that the researchers conducted in-depth interviews through qualitative analysis.

∙        As sustainable forest management, it is considered a meaningful study on the intersection of timber harvesting and forest-based ecotourism.

∙        Although the research paper shows the researcher's enthusiasm, it needs to be more organized and smoothly presented. Please refer to the suggestions for modification.

<Contents and Logical flow>

∙        Chapter 2, which is about research methods, goes beyond explaining the background and target area of the study. It seems like the research results and the content does not fit the method chapter.

∙        The method chapter should only have an objective introduction to the materials and method of this study, namely the target area and how the research was conducted.

∙        In particular, sections “2.2.1. A unique forest remnant and zoning of functions” and “2.2.2. Disputes over the BiaÅ‚owieża Forest massif’s values in Poland and Belarus” are too long and not relevant to the method. We suggest adding a new chapter between Chapter 1 Introduction and Chapter 2 Method, OR summarizing and organizing the content, excluding the result graphs, and adding them to the discussion chapter as the author's explanation and argument for the background of the study area.

∙        Table 2 is already considered as the researcher's results.

∙        Chapter 2.3 combines research methods and results, and please modify it.

∙        The author mentioned qualitative research in the abstract, but there are no narrative qualitative analysis results in Chapter 3 Results, and they are partially indicated in Chapter 4 Discussion.

∙        If the qualitative analysis has been conducted, it is appropriate to include qualitative analysis results in Chapter 3 Research Results using professional analysis tools rather than presenting them as they are in the discussion of Chapter 4.

<Format and arrangement>

∙        Readers have no information about the target area. To help readers understand, please add four field photos in Chapter 4. Maps are insufficient to understand the field.

∙        Please redesign Figure 1. The letters appear blurry. Please make it a table, not a picture.

∙        In Figure 2, please show what the black and white colors represent and what they mean. Also, if the author did not create the Figure 2 map, please indicate the source.

∙        Figure 3 is more like a table than a figure, and the table format is incorrect. Please make the necessary corrections.

∙        The legend of Figure 4 is unreadable. Please increase the font size and make it clear.

∙        Figure 5 is more like a table than a figure.

∙        Figure 6; The text explaining the horizontal and vertical axes of each graph in Figure 6 is too small. Also, it looks awkward to have an outline around the entire Figure 6. Please make the necessary corrections.

Reviewer 3 Report

This study is an intriguing and pertinent insight into park conservation. The comparison of two cultures and economic interests highlights valuable sociological and psychological factors for management practices. The English is used correctly but is a bit stilted. The abstract should be edited to make it more flowing and comprehensible to non-scientists, who are likely to be the decision makers in areas such as this.

Round 2

Reviewer 1 Report

The authors have satisfactorily addressed my comments on the earlier version.

Author Response

Thank you for your satisfaction with our revised version.

Reviewer 2 Report

We express our gratitude to the authors for their diligent efforts in revising the article. We appreciate your faithful and thorough response to the lengthy review and your genuine reply.

Author Response

(The authors gave the same response as above.)
